# Directed self-assembly of liquid crystalline blue-phases into ideal single-crystals

Jose A. Martínez-González[1,2,*], Xiao Li[1,*], Monirosadat Sadati[1], Ye Zhou[1], Rui Zhang[1], Paul F. Nealey[1,2] & Juan J. de Pablo[1,2]

Chiral nematic liquid crystals are known to form blue phases—liquid states of matter that exhibit ordered cubic arrangements of topological defects. Blue-phase specimens, however, are generally polycrystalline, consisting of randomly oriented domains that limit their performance in applications. A strategy that relies on nano-patterned substrates is presented here for preparation of stable, macroscopic single-crystal blue-phase materials. Different template designs are conceived to exert control over different planes of the blue-phase lattice orientation with respect to the underlying substrate. Experiments are then used to demonstrate that it is indeed possible to create stable single-crystal blue-phase domains with the desired orientation over large regions. These results provide a potential avenue to fully exploit the electro-optical properties of blue phases, which have been hindered by the existence of grain boundaries.

[1] Institute for Molecular Engineering, The University of Chicago, 5640 South Ellis Avenue, Chicago, Illinois 60637, USA. [2] Material Science Division, Argonne National Laboratory, Lemont, Illinois 60439, USA. * These authors contributed equally to this work. Correspondence and requests for materials should be addressed to P.F.N. (email: nealey@uchicago.edu) or to J.J.d.P. (email: depablo@uchicago.edu).

**B**lue phases (BPs) represent chiral liquid-crystalline states where molecules spontaneously form structures consisting of double-twisted cylinders. Such cylinders then adopt crystalline arrangements that are ultimately responsible for the materials' colours. BPs appear in a narrow range of temperature, between the isotropic (I) and cholesteric (Chol) states[1,2]. In the so-called BPI and BPII, the double-twisted cylinders are arranged in a cubic crystalline structure with a body centre cubic or a simple cubic symmetry, respectively. Such structures are accompanied by the formation of ordered networks of topological defects that reflect light in the visible range. A third BP, the so-called BPIII, exhibits a disordered structure[3,4]. The highly ordered morphology of BPI and BPII gives rise to unusual physical properties, including a high viscosity, Bragg reflection of visible light, a finite shear modulus and a fast optical response (much faster than that of traditional nematic liquid crystals). These properties are desirable for technologies involving photonic materials[5–9], electro-optical devices[7,10] and biological sensors[11,12]. Applications of BPs to date have been limited by two important shortcomings. First, BPs only arise over a narrow range of temperature, approximately $\Delta T \sim 1 \,^\circ$C. Second, BP specimens are generally polycrystalline, consisting of many small multi-platelet domains, each one reflecting light according to its orientation, which affects the intensity of the Bragg reflections and the operating voltage[13–15]. Recent work has shown that it is possible to increase the thermal stability of BP's by polymerization[16–20], by inclusion of colloidal nanoparticles[21–25] or by confinement into micrometre-sized droplets[12]. Recent studies also indicate that by relying on electric, thermal or surface treatments[15,26–30], it is possible to produce BP monodomains, which represent polycrystalline specimens where small crystalline platelets ($\sim 10 \,\mu$m) have the same lattice plane parallel to the substrate[15,26,27]. Although the grain boundaries between platelets interfere with performance, such specimens have been shown to have superior characteristics to those of conventional BPs, including better electro-optical properties (for example, a low operating voltage), high transmission and reduced hysteresis[13,14]. Importantly, it has not been possible to create macroscopic specimens of ideal, single-crystal BPs with a specific crystallographic plane orientation.

The lattice orientation of a BP is mediated by the strain induced by any confining geometry and the corresponding anchoring conditions; parallel anchoring at a surface or interface induces molecules to lay down on that surface, and homeotropic anchoring causes them to adopt a perpendicular orientation.

In this work, theory and simulations are used to design patterned surfaces that combine homeotropic and planar anchoring regions that are capable of directing and stabilizing the lattice orientation of BPs. Such surfaces, which we prepare by grafting polymer brushes to select regions of a patterned substrate, are then shown to enable formation of single-crystal BPs over macroscopic regions. More specifically, we rely on a Landau-de Gennes theory for the tensor order parameter and continuum simulations to account for the effects of enthalpic, elastic and surface contributions to the free energy[12,31–34]. Our experiments are carried out on 3.5 $\mu$m-thick films deposited on lithographically nano-patterned surfaces. We focus on BPII, whose unit cell exhibits a simple cubic structure that reflects light along all crystallographic planes. However, some results are also provided for BPI.

## Results

**Pattern templates.** The overall aim of this work is to determine whether proper pattern symmetries exist that can lead to formation of a single-crystal, ordered BP (BPI or BPII) having a specific crystallographic orientation. A first line of thinking could be to construct patterns that follow the preferred molecular alignment of the liquid crystal at different crystallographic planes of a BP in the bulk (see Supplementary Fig. 1). Such an alignment, however, would depend on where the crystallographic plane is cut (see Supplementary Fig. 2). The design of surface patterns based on this information would introduce additional strain on the BP structure if the film's thickness was not commensurate with the BP-lattice constant. Moreover, confining the BP into a film would break the symmetry in the vicinity of the boundaries, even for weak anchoring conditions[12]. To address these challenges, we therefore begin by analysing how a BP responds to confinement.

The BPII crystallographic planes are denoted by $(hkl)$, where $h$, $k$ and $l$ are the Miller indices. In this work BPII$_{(hkl)}$ denotes a BPII oriented with the $(hkl)$ plane parallel to the surface. For light that is incident normal to a $(hkl)$ plane, the reflected light has wavelength

$$\lambda_{[hkl]} = \frac{2n\,a}{\sqrt{h^2 + k^2 + l^2}},$$

where $n$ is the diffraction index and $a$ is the unit cell size. Experimentally, Bukusoglu et al.[11] found that a BPII film with lattice size of 150 nm, under uniform planar or homeotropic conditions, is polycrystalline and is dominated by BPII$_{(111)}$ domains inter-dispersed by a few BPII$_{(110)}$ and BPII$_{(100)}$ platelets[9]. Such monodomains reflect light with $\lambda_{(111)} = 260$ nm, $\lambda_{(110)} = 318$ nm and $\lambda_{(100)} = 450$ nm. The structural stability of a BP depends on the interplay between surface and bulk contributions to the free energy; identifying conditions that favour a given BPII-lattice orientation with respect to a surface therefore requires an analysis of the system's response to confinement. To that end, building on results reported in the literature[11,12], we carried out numerical simulations of a typical BPII with unit cell size $a_{\mathrm{BPII}} = 150$ nm, confined into a 2 $\mu$m-thick channel with homeotropic anchoring at the top and bottom surfaces (additional details are provided in the Methods section).

Through careful choice of the initial conditions, one can generate three monocrystalline reference states, namely BPII$_{(100)}$, BPII$_{(110)}$ and BPII$_{(111)}$ (see the Methods section for additional details). Figure 1 shows the corresponding topological defects and the average molecular orientation in the immediate vicinity of the homeotropic surface, along with a two-dimensional map of the behaviour of the scalar order parameter (S), which measures the local degree of molecular ordering, evaluated immediate vicinity of the surface. For each of the reference states, BPII$_{(100)}$, BPII$_{(110)}$ and BPII$_{(111)}$, there is a correlation between the symmetry of the S-maps and the preferred molecular alignment: a value of $S = 1$ corresponds to a material that is perfectly homeotropic at the surface, whereas $S \approx 0$ corresponds to an isotropic, disordered region. The S-map allows one to identify which regions near the surface undergo costly elastic distortions as a result of the tendency of a material to adopt an average orientation that is different from that imposed by the surface anchoring. For instance, topological defects—which result from abrupt changes of the local molecular order and where there is no preferred molecular orientation—appear in the S-map as regions where an abrupt change of colour occurs; in the bulk, such abrupt changes correspond to topological line defects that can be represented as the blue isosurfaces with $S = 0.35$ shown in Fig. 1. Each BPII presents a near-surface S-map that is characteristic for a particular crystallographic orientation. By recognizing that, to first order, surface-order maps provide an indication of the local strain (and therefore energetic cost) associated with presenting a particular crystallographic plane onto a surface, such maps provide a blueprint for creating patterns of planar and homeotropic regions whose aim is to relieve the elastic distortions

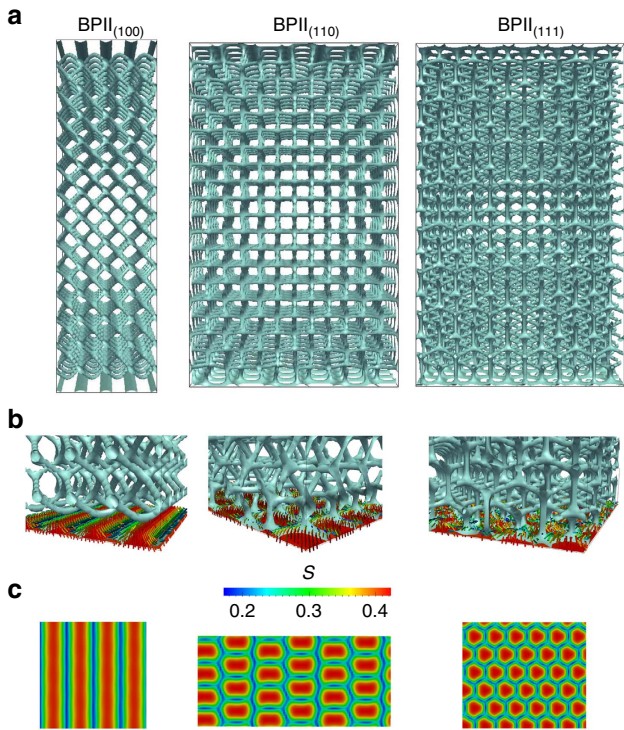

**Figure 1 | Defect structure and scalar order parameter maps for BPII.** (**a**) Defect structure of the BPII with different lattice orientations. (**b**) Close-up of the BPII-topological line defects and molecular orientation in the proximity of uniform homeotropic interfaces. (**c**) Scalar order parameter maps at the corresponding interfaces. To represent low and high values of $S$ we use colour maps that go from blue (disordered) to red (ordered). In the case of the local director, the colour corresponds to the projection of the molecular orientation onto the surface normal vector; as a result, blue directors are parallel to the surface, while red directors are perpendicular to the surface.

induced by the surface. Moreover, the symmetry of the S-map appears to be essentially unaffected when the channel's thickness varies (see Supplementary Figs 3–5). Building on this idea, in Fig. 2 we show the particular designs that we hypothesize will direct the BPII orientation along specific Miller indices.

**Optimal pattern designs.** The stripe-like pattern of Fig. 2a was conceived to produce monocrystalline $BPII_{(100)}$ domains that reflect visible light with $\lambda_{(100)} = 450$ nm. We found from simulations using a Landau-de Gennes formalism (see Methods section) that the periodicity of the stripes is equal to the BPII-lattice constant; therefore, for the stripe-pattern design, the sum of the planar, $P$, and homeotropic, $H$, stripe widths, must be equal to the BP lattice size (as indicated in Fig. 2a). We use the homeotropic ratio, $HR = H/(P + H)$, to determine the optimal proportion of planar and homeotropic contributions from the pattern. Figure 3a shows the free energy density difference between the confined system and its value in the bulk, $\Delta f$, as a function of HR for $P + H = a_{BPII} = 150$ nm. Although $BPII_{(111)}$ is the most favourable configuration for either uniform planar (HR = 0) or uniform homeotropic (HR = 1) anchoring, the Stripe pattern reverses that trend by reducing the free energy of the $BPII_{(100)}$, which becomes the most stable orientation for $0.2 < HR < 0.8$. A value of $HR = 0.5$ is optimal for Stripe patterns.

The patterns that we propose to stabilize $BPII_{(110)}$ and $BPII_{(111)}$ consist of hexagonal arrays of rectangles or circles of homeotropic

anchoring over a planar background. In what follows we refer to them as rectangular and circular patterns, respectively (Fig. 2b,c). We find that the parameters associated with the spatial distribution of these homeotropic regions depend on the size of the BPII unit cell, $a_{BPII}$, and the lattice orientation (see Fig. 2). For these nano-patterns, the area of the rectangles is $2W \times W$ and the radius of the circular domains is denoted by $r$; by changing $W$ and $r$ in a systematic manner, one can determine the optimal conditions to produce single crystals of $BPII_{(110)}$ and $BPII_{(111)}$, respectively. Figure 3b, shows the $W$-range over which $BPII_{(110)}$ becomes the stable configuration when the blue-phase is confined into a film on a rectangular patterned surface. In the case of the circular pattern, $BPII_{(111)}$ is the stable state for all values of $r$ considered here, as shown in Fig. 3c. It is important to note that, due to the strong anchoring conditions imposed by the pattern regions, the surface energies associated with different BP orientations are similar for all three cases considered above (Fig. 3d–f); the overall behaviour of the free energy and the changes induced by the patterns are primarily due to elastic distortions. As seen in Fig. 3, the energetic cost associated with such distortions is minimal and represents only a small fraction of the overall energy of the system. It is, however, sufficient to influence the orientation of the entire material over macroscopic regions. The main role of the patterned surface is therefore that of seeding the correct crystallographic orientation in the proximity of the interface, and favour the (100), (110) and (111) lattice orientations with stripes, rectangles and circles, respectively; these shapes and the corresponding surface anchoring induce a local deformation that prevents the material from adopting other, more unfavourable orientations (see Fig. 4). Based on these theoretical results, it is reasonable to infer that the patterned surfaces shown in Fig. 2 will direct the orientation of a BPII film. These predictions are examined in the following section.

**Experimental results.** To summarize our design strategy, binary homeotropic/planar patterns are first designed from continuum simulations of a $BPII_{(hkl)}$ under uniform homeotropic interfacial conditions. The S-maps at the interface are correlated with the preferred molecular alignment: BP molecules in regions with the highest order parameter (red sections) show a preferred perpendicular alignment at the interface. In the other regions, the preferred molecular alignment deviates slightly from that imposed by the interface and is associated with a preference for planar alignment, as indicated by the behaviour of the director field above the interface. The S-maps are simplified into a binary pattern consisting of planar and homeotropic regions; the symmetry of the patterns is described in terms of the BP lattice constant. Landau-de Gennes calculations are then performed to determine the optimal dimensions of the homeotropic and planar regions (that is, the optimal values of HR, $W$ and $r$ of the striped, rectangular and circular patterns, respectively). Once a pattern is optimized, the resulting information can be used to experimentally prepare each pattern of interest (see Supplementary Fig. 6).

Experiments were carried out on a silicone substrate with an approximately 5 nm-thick synthetic grafted polymer brush that imposes homeotropic anchoring on the planar substrate. Following Li et al.[35], nanopatterns were produced through a lithographic process using e-beam on a polymer-covered surface (see Supplementary Fig. 7). The technique of Li et al.[35] is particularly helpful in that it enables preparation of flat patterned surfaces that are devoid of micrometre-scale topographic steps. The chiral liquid crystals considered here were prepared by mixing the mesogen MLC 2142 with 36.3 wt% of the chiral dopant 4-(1-methylhepty-loxycarbonyl)phenyl-4-hexyloxybenzonate (S-811). This mixture

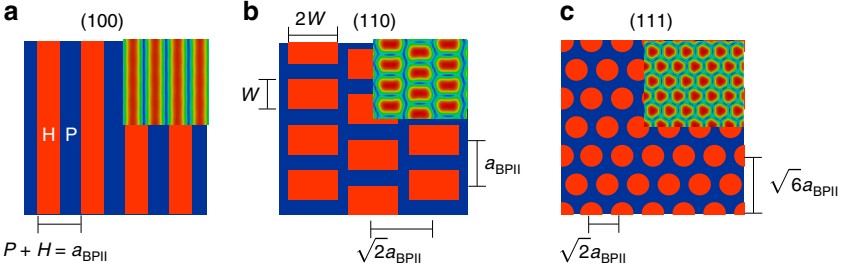

**Figure 2 | Pattern templates for BPII with different lattice orientations.** (**a**) Stripe pattern for BPII$_{(100)}$; (**b**) rectangular-array pattern for BPII$_{(110)}$; (**c**) circular-array pattern for BPII$_{(111)}$. Insets correspond to the $S$-map, the symmetry and spatial dimensions of these maps are simplified into binary patterns where red and blue sections correspond to homeotropic and planar regions, respectively. Colour map as in Fig. 1.

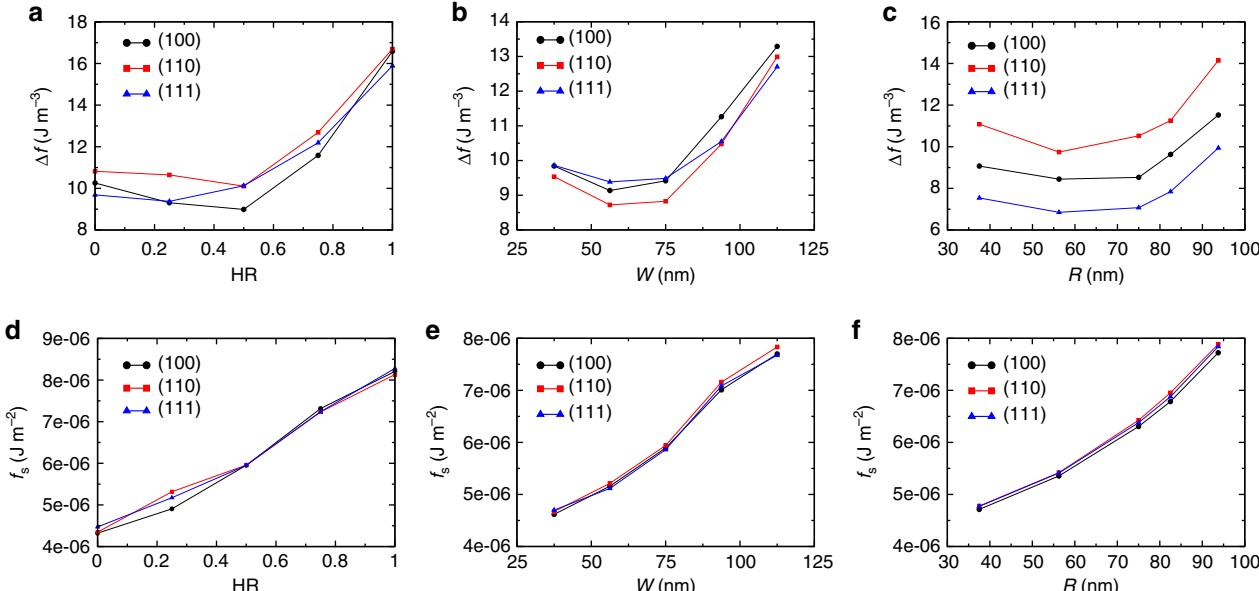

**Figure 3 | Free-energy densities of BPII on different patterned surfaces.** (**a–c**) Free-energy density difference, $\Delta f = f - f_{Bulk}$, and (**d–f**) surface free energy density, $f_S$, as a function of different pattern parameters, for different BPII-lattice orientations: BPII$_{(100)}$ (circles), BPII$_{(110)}$ (squares) and BPII$_{(111)}$ (triangles). Consistent with experimental observations, BPII$_{(111)}$ is obtained for uniform homeotropic and planar anchoring (HR = 1 and HR = 0, of the stripe-like pattern). These results show the parameter intervals over which each pattern favours a particular lattice orientation; thus BPII$_{(100)}$ is favoured when HR ≈ 0.5, BPII$_{(110)}$ when $W$ ≈ 75 nm and BPII$_{(111)}$ for all values of $r$.

produces a BPI and a BPII with lattice sizes $a_{BPI} \approx 255$ nm and $a_{BPII} \approx 150$ nm, respectively[11].

This liquid crystal mixture was confined into 3.5 µm-thick slits with homeotropic anchoring on the top surface and a 0.25 mm² patterned area on the bottom surface. To provide a reference for the influence of the patterns, in all samples the patterned area was surrounded by a region of uniform homeotropic anchoring.

Following the predictions outlined above, for the Stripe patterns we use HR = 0.5 and $P + H = a_{BPII}$; for the rectangular and circular patterns we use $2W = 2r = a_{BPII}$. In this way, all the patterns are produced in terms of the unit cell size, a feature that will be useful for extending the results presented here to systems having different chirality.

Figure 5a shows optical micrographs of the experimental systems. As one can see from the figure, the patterned surfaces do not significantly affect the structural and thermal behaviour of the Chol and BPI phases. For BPII, however, our results demonstrate that patterns induce the formation of a single-crystal BPII specimen over the entire patterned area, with no platelet-like domains or grain boundaries. In all cases, the micrographs were taken for light normally incident in the reflection mode of a cross

polarizer. As predicted, the stripe pattern (Fig. 5a top) induces a BPII$_{(100)}$ single crystal, which reflects light with wavelength $\lambda_{(100)} \approx 450$ nm; the rectangular and circular patterns (Fig. 5a centre and bottom) give rise to BPII$_{(110)}$ and BPII$_{(111)}$ single crystals, respectively. They appear black in the images because the reflected light in this case is outside the visible spectrum ($\lambda_{(110)} \approx 318$ nm, $\lambda_{(111)} \approx 260$ nm). Kossel diagrams were obtained using monochromatic light with $\lambda = 405$ nm; the lines shown in these diagrams correspond to light reflected by the (100) planes and reveal the lattice orientation of the BP. Our measurements are consistent with theoretically and experimentally determined Kossel diagrams[36–38], and confirm the existence of the (100) and the (110) lattice orientations on the stripe and rectangular patterns, respectively. For the circular pattern, the symmetry of the diagram can be explained by analysing the lattice structure of the BPII$_{(111)}$ in the proximity of the patterned surface. We find that such a structure depends on the channel thickness and is consistent with the formation of a hexagonal BP layer at the wall, as revealed by the symmetry of line defects (see Supplementary Fig. 8). As the name indicates, the hexagonal BP consists of a hexagonal array of double twist

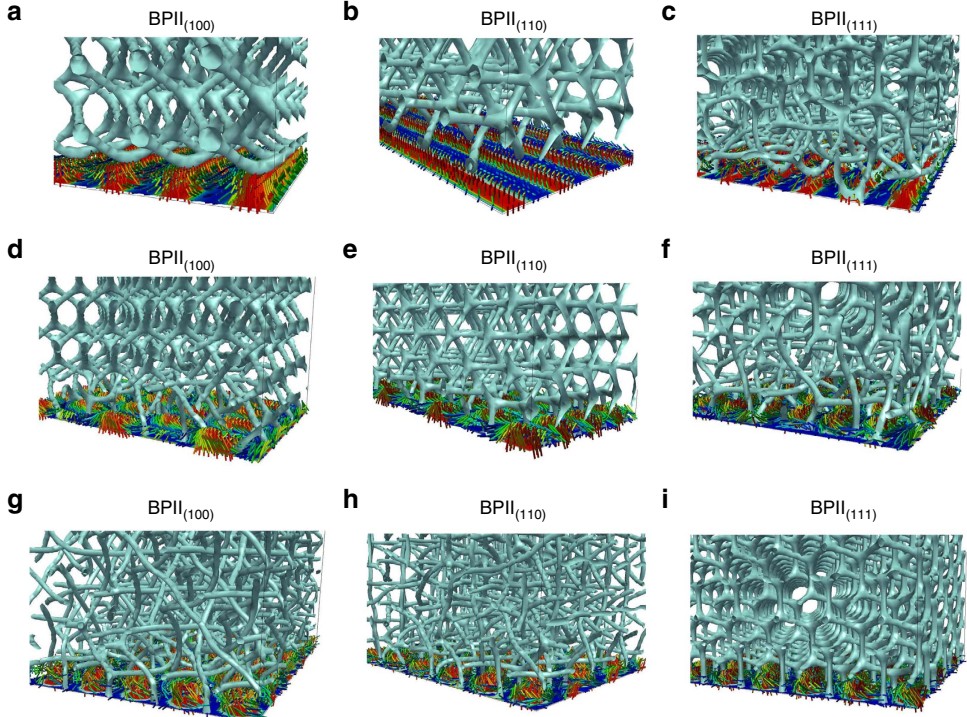

**Figure 4 | BPII topological defects in the proximity of patterned surfaces.** Close up of the defects and molecular orientation in the proximity of the stripe (**a–c**), rectangular (**d–f**) and circular (**g–i**) patterned surfaces for (100), (110) and (111)-BPII lattice orientations. Colour map as in Fig. 1.

cylinders; the corresponding Kossel diagram agrees with that shown in Fig. 5 for the circular pattern[39,40]. Our simulation results, shown in the Supplementary Fig. 8, show that the circular pattern produces a BPII$_{(111)}$ monodomain where the lattice structure adopts a hexagonal symmetry in the vicinity of the patterned surface, and this structure is revealed experimentally by the Kossel diagram.

Additional experimental evidence for the single-crystal characteristic of the domains produced by the process outlined here can also be obtained from the Kossel diagrams. In a single crystal, Kossel diagrams obtained from different regions of a sample should be identical. In contrast, in a polycrystalline monodomain, platelets having different $x$–$y$ orientations would produce Kossel diagrams that differ in their relative orientation. This is exactly what is found in our experiments, as shown in Fig. 5b,c. Kossel diagrams extracted from different regions of the single crystals produced here have the same orientation, whereas those extracted from polycrystalline monodomains do not (additional results can be found in the Supplementary Fig. 9).

## Discussion

As explained above, the pattern templates proposed here were conceived on the basis of theoretical results from a tensorial description of the preferred local molecular orientation of the chiral liquid crystal in the proximity of a planar or homeotropic interface—S-maps at planar interfaces where ruled out in this work for practical reasons, as they exhibit more complex symmetries (see Supplementary Fig. 10). The order parameter is able to identify and report regions of high elastic distortions, which a surface pattern is able to relieve. The preferred orientation of the BP is thus mediated by the strain imposed by a patterned surface, and the generality of this concept is established by designing different patterns that stabilize different orientations along the BPII$_{(100)}$, BPII$_{(110)}$ and BPII$_{(111)}$

crystallographic lattice planes. In this context, we hasten to point out that Cattaneo *et al.*[41] have shown that square-like patchy patterns can be used to induce formation of Skyrmion-like phases in non-chiral liquid crystals—the molecules of this phases form chiral twists but the entire phase does not possess a crystalline symmetry. A closer inspection of the LC texture in the proximity of our BPII$_{(111)}$ and BPII$_{(110)}$ interfaces, also reveals the formation of Skyrmion structures reminiscent of those observed by these authors (see Supplementary Fig. 11); similarly, the patterns that we propose, made of hexagonal arrangements of rectangular and circular sections, are connected with the natural Skyrmion-like behaviour of the molecules at those BP lattice orientations.

The strategy presented here to create single-crystals appears to be general, and can also be applied to other phases, including the BPI. Figure 6a summarizes the process for obtaining a single-crystal of the BPI$_{(110)}$, where the BP lattice parameter is $a_{BPI} = 260$ nm. The pattern in this case consists of a hexagonal arrangement of rectangles. Figure 7, shows the reflection spectrum of the Chol phase and the polycrystalline and single-crystal domains of BPI and BPII, respectively. The Chol phase exhibits a constant intensity value of reflected light, whereas BPI and BPII show highly selective light reflection according to their crystalline structure. As can be appreciated from Fig. 7, single-crystal BP domains exhibit a marked improvement in the intensity of reflected light when compared with polycrystalline samples. Polycrystalline domains give rise to spectra with multiple peaks, which are a consequence of the non-uniform lattice orientation of different platelets in the samples; the spectra for single-crystal domains exhibit a single, sharp peak, which is consistent with the uniformity of the lattice orientation in the entire sample.

The approach proposed in this work for directed assembly of single-crystal BPs provides a platform for development of devices that might permit full exploitation of the structural and electro-optical properties of BPs. By removing polydomain

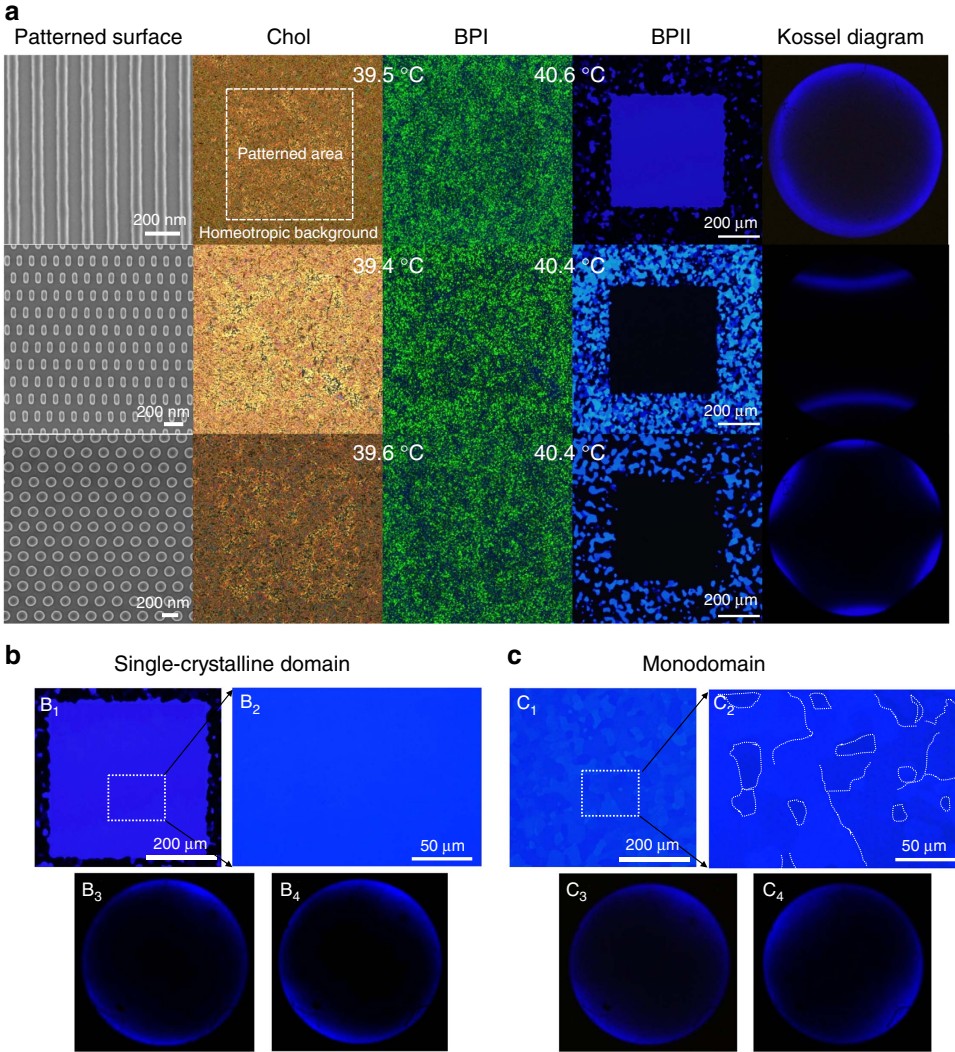

**Figure 5 | Experimental confirmation of directed BP-lattice orientation by patterned surfaces. (a)** Scanning electron microscopy images of three different patterned surfaces, along with the corresponding micrographs of the cholesteric and BPs I and II and the Kossel diagrams of the BPII in the patterned area. The Chol-BPI and BPI-BPII transition temperatures for each case are also indicated. As predicted by our theoretical results, each pattern stabilizes the BPII orientation for that it was designed to stabilize. The resulting single-crystal monodomains are as large as the patterned area. **(b,c)** Reflected light optical images of the BP cell under crossed polarizers: **(b)** single crystalline domain on stripe patterned surface; **(c)** monodomain adjacent to the stripe pattern area. The Kossel diagrams correspond to the right and left sides of the zoomed-in images.

structures and grain boundaries, key properties of an optical device, such as its transmission and response time, can be enhanced significantly. As the technique employed here to prepare chemically-patterned surfaces has been previously used to pattern areas as large as 300 mm wafer[42], it offers the potential to grow single-crystal BPs specimens of such dimensions. Furthermore, by relying on the perfect defect networks presented by single-crystal BPs, it might be possible to template the assembly or synthesis of polymeric meshes for applications in sensing and separations, liquid crystal lasers with a desired bandgap, and mechano-photonic metamaterials.

## Methods

**Landau-de gennes modelling.** The free energy, $F$, of the chiral liquid crystal considered in this work was described in terms of a continuum mean field Landau-de Gennes formalism[29–32]. In this model $F = F(Q)$, where $Q$ is the tensor order parameter, defined by $Q_{ij} = S (n_i n_j - 1/3 \, \delta_{ij})$. Here $i,j = 1, 2, 3$ and $n_i$ are de $x, y, z$ components of the local director vector; $S$ is the scalar order parameter, given by $S \leq 3/2 \cos^2\theta - 1/2 >$, with $\cos \theta = a \cdot n$, where $a$ is the molecular orientation

and $< >$ denotes a spatial average. Thus, the tensor order parameter contains the structural information of the liquid crystalline phase.

The free-energy functional accounts for short-range ($f_P$), long-range elastic ($f_E$) and surface ($f_S$) contributions, that is,

$$F(Q) = \int d^3x[f_P(Q) + f_E(Q)] + \int d^2x f_S(Q), \qquad (1)$$

where the short-range contribution is given by

$$f_P = \frac{A}{2}\left(1 - \frac{U}{3}\right)\text{tr}(Q^2) - \frac{AU}{3}\text{tr}(Q^3) + \frac{AU}{4}\text{tr}(Q^2)^2. \qquad (2)$$

In equation (2), $A$ and $U$ are phenomenological parameters that depend on temperature and pressure[31]. The long-range elastic contributions to the free energy are given by[41,43],

$$f_E = \frac{1}{2}\left[L_1\frac{\partial Q_{ij}}{\partial x_k}\frac{\partial Q_{ij}}{\partial x_k} + L_2\frac{\partial Q_{jk}}{\partial x_k}\frac{\partial Q_{jl}}{\partial x_l} + L_3 Q_{ij}\frac{\partial Q_{kl}}{\partial x_i}\frac{\partial Q_{kl}}{\partial x_j} + L_4\frac{\partial Q_{jk}}{\partial x_l}\frac{\partial Q_{jl}}{\partial x_k} + L_5 2q_0\epsilon_{ikl}Q_{ij}\frac{\partial Q_{lj}}{\partial x_k}\right], \qquad (3)$$

where $\varepsilon_{ikl}$ is the Levi–Civita tensor, $q_0 = 2\pi/p$ is the inverse of the pitch and measures the chirality of the system. $L_i$ is the elastic constant of the liquid crystal. The last term of equation (1) corresponds to the surface contributions to the free energy. The patterned surfaces considered in this work consist of regions with planar and homeotropic anchoring. In the first case, a molecular orientation parallel to the surface

is imposed, but without any preferential direction on the plane; this condition is referred to as planar degenerate anchoring, and the corresponding free energy is given by[44],

$$f_S^P = W_P(\tilde{Q} - \tilde{Q}^\perp)^2 + W_P(\tilde{Q} : \tilde{Q} - S_0^2)^2, \tag{4}$$

where $S_0$ is the surface preferred degree of order, $W_P$ is the planar anchoring energy, $\tilde{Q} = Q + SI/3$ and $\tilde{Q}^\perp = P\tilde{Q}P$, $P$ is the projection operator $P_{ij} = \delta_{ij} - v_i v_j$ and $v$ is the vector normal to the surface. For the homeotropic region, surface contributions to the free energy are given by,

$$f_S^H = \frac{1}{2} W_H (Q - Q^0)^2, \tag{5}$$

where $W_H$ is the homeotropic anchoring energy and $Q^0$ is a surface-preference tensor order parameter.

Stable and metastable states were found by minimization of the free energy; this was achieved by means of a Ginzburg–Landau relaxation method where $Q$ evolves towards equilibrium according to[32,33],

$$\frac{\partial Q}{\partial t} = -\frac{1}{\gamma}\left[\Pi\left(\frac{\delta F}{\delta Q}\right)\right], \tag{6}$$

with boundary conditions such that $\Pi[(\delta F/\delta\nabla Q)\cdot v]=0$. Parameter $\gamma$ represents a diffusion coefficient and the operator $\Pi(B)=1/2(B+B^T)-1/3\mathrm{tr}(B)I$ ensures the symmetric and traceless properties of the $Q$-tensor parameter.

Initial configurations for BPI and BPII were generated as follows[29]: for BPI:

$$Q_{xx} = A\left(-\sin\left(ky/\sqrt{2}\right)\cos\left(kx/\sqrt{2}\right) - \sin\left(kx/\sqrt{2}\right)\cos\left(kz/\sqrt{2}\right) + 2\sin\left(kz/\sqrt{2}\right)\cos\left(ky/\sqrt{2}\right)\right) \tag{7}$$

$$Q_{xy} = A\left(-\sqrt{2}\sin\left(kx/\sqrt{2}\right)\sin\left(kz/\sqrt{2}\right) - \sqrt{2}\cos\left(ky/\sqrt{2}\right)\cos\left(kz/\sqrt{2}\right) + \sin\left(kx/\sqrt{2}\right)\cos\left(ky/\sqrt{2}\right)\right) \tag{8}$$

For BPII:

$$Q_{xx} = A(\cos kz - \cos ky) \tag{9}$$

$$Q_{xy} = A\sin kz, \tag{10}$$

where the strength of the chirality is given by $k = 2q_0 r$ and $r$ is the redshift, which was found to be 0.71 for BPI and 0.86 for BPII. The amplitude of initialization is $A = 0.2$. The BP lattice parameters for BPI ($a_{BPI}$) and BPII ($a_{BPII}$) are related to the chiral pitch and the red shift as follows: $a_{BPI} = \frac{p}{\sqrt{2}r}$, $a_{BPII} = \frac{p}{2r}$. In all cases, the components $yy$, $zz$, $xz$ and $yz$ were obtained by cyclic permutation of those given above.

For the description of the system we use a lattice array with a mesh resolution of 7.5 nm and typical values of anchoring energies, that is, $W_P = 4 \times 10^{-3}$ Jm$^{-2}$ and $W_H = 8 \times 10^{-4}$ Jm$^{-2}$; for the chiral liquid, we use the following values, which were found to match experimental observations[11,12], A $= 1.067 \times 10^5$ Jm$^{-3}$, $L_1 = 6$ pN and $L_5 = 2L_1$. For BPs we consider a chiral pitch of 258 nm, $U = 2.755$ for BPII and $U = 3.0$ for BPI. For simulations, we use the elastic constant, $L_1$, the coherence length, $\xi_C = \sqrt{L_1/A}$ and the extrapolation length, $\xi_S = L_1/W$, to reduce variables as follows: $r^* = r/\xi_C$; $W^* = \xi_C/\xi_S$; $F^* = F/(L_1\xi)$. The reduced temperature, $\tau$, is related to the $U$ parameter through $\tau = 9(3 - U)/U \propto (T - T^*)$, where $T^*$ is the isotropic-cholesteric transition temperature whose value depends on the material. BP topological defects were visualized as isosurfaces of the scalar order parameter with $S = 0.35$ for BPII and $S = 0.42$ for BPI (Supplementary Fig. 1).

To compare the stability of BPs having different lattice orientations with respect to the patterned surface, we considered proper initial conditions based on the ansatze of the corresponding structures. The spatial dimensions, $L_x$ and $L_y$, of the simulation box must necessarily depend on the orientation of the BP. Specifically, for a BPII oriented with the (100)-plane parallel to the surface (BPII$_{(100)}$), we used a lattice array where $L_x$ and $L_y$ are multiples of the BPII-unit cell $a$; as the unit cell lattice parameter is $a = 150$ nm, we considered a simulation box with $L_x = L_y = 600$ nm. For the BPII$_{(110)}$ case, we rotated the phase respect to the $x$-direction keeping $L_x = 600$ nm but changing $L_y$ to a multiple value of $2^{1/2}a$; we chose $L_y = 1275$ nm, which corresponds to approximately 6.01 lattices with the (110) plane parallel to the surface. The channel thickness was kept at $L_z = 2100$ nm. Special attention must be paid to the system's dimensions to properly describe the material; failure to do so leads to distorted structures that have little resemblance to the system's actual behaviour.

After the minimization process, free-energy densities corresponding to different BP orientations for a given patterned surface were compared by taking as a reference the free energy of the BP in the bulk, that is, under periodic boundary conditions.

**Chemically prepared patterned surfaces.** The 36.3 wt% S-811 in MLC 2142 mixtures were prepared by using toluene as a co-solvent. After mixing with an ultrasonic cleaner, toluene was evaporated overnight under vacuum at 60 °C.

A 4–5 nm thick poly(6-(4-methoxy-azobenzene-4′-oxy) hexyl methacrylate) (PMMAZO) film was deposited on an oxygen-plasma cleaned silicon substrate and annealed at 250 °C for 5 min under vacuum. Non-grafted PMMAZO was removed by sonication in chlorobenzene, and the remaining PMMAZO brush was found to be around 4.5 nm thick.

A 40 nm-thick GL2000 photoresist film was deposited onto the PMMAZO brush and baked at 160 °C for 5 min. Striped patterns were exposed on the resists using electron beam lithography with the JEOL 9300FS electron-beam writer at the Center for Nanoscale Materials, Argonne National Laboratory. Exposed substrates were developed with n-amyl acetate for 15 s and rinsed with isopropyl alcohol. The resulting resist pattern was transformed onto a chemical pattern on the PMMAZO brush layer by exposing the sample to an oxygen plasma, followed by stripping the GL2000 photoresist in chlorobenzene[35].

The glass microscope slides were modified by Octadecyltrichlorosilane (OTS). The OTS glass and the Si substrate with the PMMAZO chemical patterns were

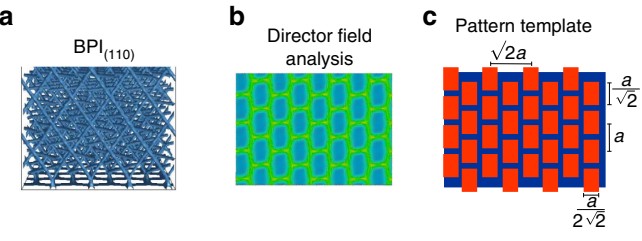

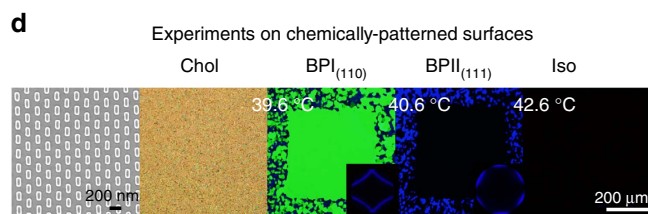

**Figure 6 | Direct self-assembly of a BPI$_{(110)}$ single crystal.** The strategy presented in this work can also be used to produce a BPI$_{(110)}$ monocrystalline domain. (**a**) Numerical simulations of a confined BPI with (110)-lattice orientation; (**b**) director field and S-map analysis; (**c**) determination of the binary pattern template; (**d**) fabrication of the chemically patterned surface and experiments to obtain the desired BPI single crystal.

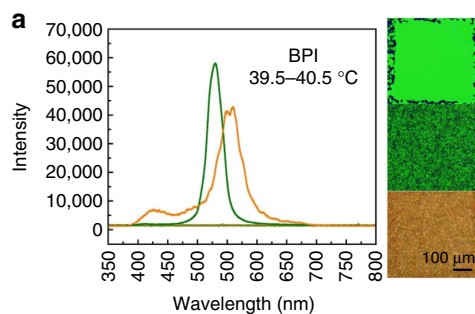

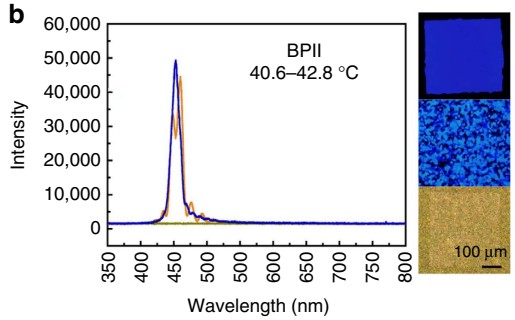

**Figure 7 | Reflection spectrum of polycrystalline and single crystal BPs.** Comparison between the reflection spectrum of Chol and polycrystalline and single-crystal domains of BPI (**a**) and BPII (**b**), respectively.

placed face-to-face, with a 3.5 μm spacer, to define the cell thickness. The optical cell and the LC were heated above the clearing point and S-811/MLC 2142 mixtures were injected through capillary action. The system was then slowly cooled down to room temperature (see extended Fig. 4). At this point, the sample was ready for the following thermal process: experiments were started by heating a cholesteric phase from 25 °C to 39.6 °C During this process no visible changes were observed. A heating rate of 0.5 °C per min was then used to reach 39.6 °C. At this point, we used a slower heating rate in which we changed the temperature by 0.2 °C every 3 min.

Optical characterization was performed using the cross-polarized and reflection modes of an Olympus BX60 microscope with a × 10 and a × 50 objective. Samples were heated up to the isotropic phase using Bioscience Tools TC-1-100s temperature controller controlling hot stage at a rate 0.2 °C every 3 min. Kossel diagrams were used to identify the type of BP and determine the crystal orientation.

Ultraviolet-visible spectra of BP samples were carried out using spectrometer (USB4000, Ocean Optics).

**Materials.** MLC 2142 and S-811 were purchased from Merck. Fisher Finest Premium Grade glass slides and coverslips were obtained from Fisher Scientific. OTS, chlorobenzene, isopropyl alcohol and n-amyl acetate were purchased from Sigma-Aldrich and used without further purification.

**Data availability.** The data that support the findings of this study, as well as the models used in our calculations, are available from the corresponding author upon reasonable request.

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

## Acknowledgements

This work is supported by the Department of Energy, Basic Energy Sciences, Materials Sciences and Engineering Division. The calculations reported here were performed on the University of Chicago Research Computing Center and on Blues, a high-performance computing cluster operated by the Laboratory Computing Resource Center at Argonne National Laboratory. Use of the Center for Nanoscale Materials was supported by the U.S. Department of Energy, Office of Science, Office of Basic Energy Sciences, under Contract Number DE-AC02-06CH11357. We also acknowledge use of the MRSEC Shared User Facilities at the University of Chicago (NSF DMR-1420709). We thank Dr Leonidas E. Ocola for help using the JEOL 9300FS electron-beam writer and Dr Junstin Jureller from the IBD NanoBiology Facility, University of Chicago. We also acknowledge the MRSEC Shared User Facilities at the University of Chicago (NSF DMR 1420709). We also thank Professor Juan P. Hernandez-Ortiz, Dr Mohammad Rahimi, Dr Abelardo Ramirez-Hernandez and Dr Yamil Colon for helpful discussions.

## Author contributions

J.J.d.P., J.A.M.-G., X.L. and P.F.N. designed the project; J.A.M.-G. and X.L. performed research; M.S. contributed to the experimental part; Y.Z. and R.Z. contributed to the theoretical analysis; J.J.d.P., J.A.M.-G. and X.L. wrote the manuscript.

## Additional information

**Competing interests:** The authors declare no competing financial interests.

