## [Peer Review File · Nature Communications]

Reviewers' comments:

Reviewer #1 (Remarks to the Author):

This paper describes a new strategy to create monocrystalline blue phase domains by uniquely designed nano-patterned surfaces. This is a carefully done study and the findings are of considerable interest. However, several points need clarifying and certain statements require justification. These are given below.

Nano-patterned surfaces were designed and produced based on the calculated 2D-maps of the scalar order parameter (S-map) for three orientations of BPII. However, the S-maps depend on the depth from a surface, that is, the S-maps are different where the lattice is cut. In this manuscript, only one surface is demonstrated for each orientation. The depth dependence should be mentioned.

The nano-patterned surface prepared by lithography in this study must be difficult to recognize a chirality of molecular arrangement of BP but may sense the order parameter S , spatially the position of the disclinations in BP. The periodicity of S due to the disclinations at a 2D surface is characterized by not the unit cell size but a half of it as shown in Figure 2. This means that the most fitted periodicity and arrangement at the surface patterns should be also characterized by a half of the unit cell size. My question is why the red patterns derived based on the unit cell size shown in Figure 2 are used although they are clearly larger than the S-map pattern exhibited as yellow one. Furthermore, experimental studies of the pitch dependence of BP and/or the pattern periodicity dependence of the substrate should be done and the extra data should be added to this manuscript to demonstrate more clearly the validity of the proposed mechanism.

Reviewer #2 (Remarks to the Author):

The manuscript is concerned with the assembly of macroscopic arrays of blue phases in chiral nematic liquid crystals on patterned substrates. It describes field-theoretic simulations and experiments and outlines a potential avenue to ideal single-crystal blue phases. The manuscript is well written and easy to follow, but oversells some aspects and lacks some detail in the description of the simulation methodology. Nevertheless, I acknowledge the novelty of this approach and recommend the manuscript for publication in *Nat. Comm.*, provided the following points below have been sufficiently addressed.

The authors demonstrate clearly that their approach is able to prepare stable and macroscopic thin films of perfectly aligned crystalline blue phase materials that are so far unsurpassed in size (area). But I find the title (ideal single-crystals) and some text in the main body of the manuscript rather misleading in the sense that cubic blue phases are truly three-dimensional materials.

Three-dimensional mono-crystalline blue phase I droplets in equilibrium with the isotropic phase have been reported in P.E. Claris, P. Pieranski, M. Joanicot, *Phys. Rev. Lett.* 52, 542 (1984). Ref. [1] states that faceted mono-crystals of up to 0.1-0.2 mm in size have been grown. The current film thicknesses of 3.5 micrometers or roughly 20 unit cells is an order of magnitude smaller than the typical grain sizes in polycrystalline blue phases. Hence, the authors need to point out more clearly that they are in fact growing a macroscopic thin film, and not a single ideal crystal. It remains to be seen whether their strategy works as well for thicker samples that may be called a single crystal.

Ref. [1] and [2] have gone down into the annals of blue phase research. Regarding the structure of blue phase III, there are more appropriate and up-to-date references, which should be also cited, e.g. S.S. Gandhi, M.S. Kim, J.-Y. Hwang, L.-C. Chien, *Adv. Mater.* 28, 8998-9005 (2016)

and O. Henrich, K. Stratford, M.E. Cates, D. Marenduzzo, Phys. Rev. Lett. 106, 107801 (2011).

In the section 'Pattern Templates' second paragraph the authors say '... A value of $S=1$ corresponds to a materials that is perfectly PARALLEL to the surface, whereas $S=0$ corresponds to an isotropic, disordered region ...' But further above the authors say the surface has been treated for homeotropic anchoring. So it should read 'perpendicular to the surface' rather than 'parallel' in the above sentence.

In the section 'Optimal Pattern Designs' second paragraph the authors mention that the surface energies associated with different BP orientations are similar for all three cases (striped, hexagonal rectangles and circles). Is this because they study a strong anchoring regime? Please add a short explanation to this statement.

Ref. [21] seems to be incorrect or garbled with another reference.

In the section 'Methods' I am missing a term in the surface free energy for planar degenerate anchoring. From Ref. [41] Eq. 3 it is evident that the trace of the square of \tilde{Q} has to be the square of the scalar order parameter $S_0 > 0$. This condition is not met without a second term as in Eq. 4 of Ref. [41].

This work contains a substantial amount of modelling and simulation. Whilst I agree that real-world SI units are more important, I would like to see also a parameter mapping from simulation units to real-world units or at least an overview of the simulation parameters that were used in this work.

Reviewer #3 (Remarks to the Author):

This paper reports a new strategy to produce large monodomain blue phase samples. The strategy is to prepare substrates with patterned anchoring interaction. The pattern is based on a free energy minimization strategy, and is verified by simulations. Experiments are carried out, producing samples in agreement with predictions.

The work is novel, interesting and the results will likely be useful for eventual device applications. I am not altogether happy with the manuscripts; I think the work is better than the manuscripts reflects.

Improvements need to be made both in organization and style.

Should the paper not begin with an introduction, then perhaps theory, simulations and experiment? The current description is rambling, and the language is unclear. What exactly does it mean that 'phases reflect light selectively'? What are external 'cues'? Do the authors really 'exert control over different planes'? 'anchoring... induces molecules to lie down'? 'It can be appreciated from the figure that the sum of Is equal to the lattice size' . 'Lattice orientation is mediated by the strain...' 'theory for the local orientation tensor...' And so on.

The style obscures the physics and the contributions made by the authors; this badly needs to be rectified.

There are some scientific questions that need to be addressed as well. The assertion is made that 'it has not been possible to create macroscopic specimens ..' followed by the claim that the proposed method make samples can be made over 'arbitrarily large macroscopic areas'. Some number are needed here. It is not uncommon to see domains of ~ 100 microns; the manuscript reports domains of 200 microns. I understand about control, but there needs to be some comparison of what is now available, and what is realistically possible with the new method. The second issue is thickness. I understand 2.5 microns for simulations, but 3.5 microns is thin. What are device requirements? How far does the induced structure extend into the bulk in thicker samples?

In crystallization, the rate of cooling is usually key in determining domain size. I found no mention of cooling rate, or indeed of any thermal processing, in the manuscript. Other key contributors are

thermal gradients – again not discussed.

The scalar order parameter S is not defined. $S=1$ indicates a 'material that is perfectly parallel to the surface' – what does that mean? The director is parallel? The molecules are parallel (isn't that $S=-1/2$)?

Without a clear definition, the whole business of S -maps is murky.

Again, the work is far better than the manuscript indicates. I urge the authors to strive for clarity and make revisions as indicated.

Reviewers' comments:

Reviewer #1 (Remarks to the Author):

This paper describes a new strategy to create monocrystalline blue phase domains by uniquely designed nano-patterned surfaces. This is a carefully done study and the findings are of considerable interest. However, several points need clarifying and certain statements require justification. These are given below.

Q1. *Nano-patterned surfaces were designed and produced based on the calculated 2D-maps of the scalar order parameter(S-map) for three orientations of BP11. However, the S-maps depend on the depth from a surface, that is, the S-maps are different where the lattice is cut. In this manuscript, only one surface is demonstrated for each orientation. The depth dependence should be mentioned.*

R: We thank the Referee for this important remark. Let's first consider the S-map behavior in the bulk. In this case, as the Referee points out, S-maps are different depending on where the lattice is cut. They give information about regions where the disclination lines are located, but not about the preferred molecular alignment. This is because the order parameter is a measure of the degree of molecular order, but not of the molecular orientation. We have included Figure S2 in the Supporting Information to illustrate this point.

When blue-phases are confined between homogenous homeotropic surfaces, the situation is different than in the bulk, particularly with regards to the molecular order near the interfaces. We identified a distinct correlation between the symmetry of the S-map and the preferred molecular alignment in the immediate proximity of the surface: the order parameter is higher in regions where the molecular alignment of the BP above the surface is perpendicular (in agreement with the homeotropic conditions of the surface), and it is lower in the vicinity of a disclination or when the director field exhibits a planar tendency (in these regions there is a perturbation of the homeotropic alignment associated with an increment of the strain on the BP unit cell). Therefore, S-maps of homeotropic surfaces are good indicators of the preferred molecular orientation and provide an image of the pattern symmetry that the blue phase must follow in order to reduce the interfacial strain caused by confinement. To complete our response to the Referee, we have also included results showing how the symmetry of the S-maps remains essentially unaffected by the channel thickness. Supplementary figures S3-S5 are included in the revised version to complement our explanation about the interpretation of the S-maps and their behavior for different channel thickness; in all cases, the S-maps show an improvement in the color to highlight contrast.

Following the Referee's concerns, we have modified the Pattern Templates section to include the explanation above.

Q2. *The nano-patterned surface prepared by lithography in this study must be difficult to recognize a chirality of molecular arrangement of BP but may sense the order parameter S, spatially the position of the disclinations in BP. The periodicity of S due to the disclinations at a 2D surface is characterized by not the unit cell size but a half of it as shown in Figure 2. This means that the most fitted periodicity and arrangement at the surface patterns should be also*

characterized by a half of the unit cell size. My question is why the red patterns derived based on the unit cell size shown in Figure 2 are used although they are clearly larger than the S-map pattern exhibited as yellow one.

R: We apologize to the Referee for the confusion that we caused by placing the S-maps obtained from simulations as insets in the pattern designs of Figure 2. We have included a statement in the caption of Fig 2 to indicate that patterns derived from simulations have the same spatial dimensions than the binary-pattern templates.

Q3. Furthermore, experimental studies of the pitch dependence of BP and/or the pattern periodicity dependence of the substrate should be done and the extra data should be added to this manuscript to demonstrate more clearly the validity of the proposed mechanism.

R: We thank the Referee for raising these important points. Patterns derived from simulations are presented in terms of the BP-lattice constant; for a BP-II, they depend on the chiral pitch, p , through $a_{\text{BP-II}}=p/2r_s$, where the redshift is found to be $r_s=0.86$ (as explained in the Methods section). Since a change in the chiral pitch implies a change in the pattern dimensions, but not in its symmetry, patterns with proper dimensions should work for blue-phases with different chirality. On the other hand, a systematic study of the effect of the pattern periodicity and the ratio of the planar and homeotropic anchoring regions on the BP stabilization would help identify a window that could be used to stabilize BPs with certain lattice orientations. Both of the above points deserve a separate, complete study, which we are currently preparing. Finally, we want to share with the Referee a figure showing some of these preliminary results, where the formation of BP₍₁₀₀₎ samples is achieved in Stripe patterns with different (P+H)-periodicities (for each image, the temperature is indicated).

Reviewer #2 (Remarks to the Author):

The manuscript is concerned with the assembly of macroscopic arrays of blue phases in chiral nematic liquid crystals on patterned substrates. It describes field-theoretic simulations and experiments and outlines a potential avenue to ideal single-crystal blue phases. The manuscript is well written and easy to follow, but oversells some aspects and lacks some detail in the description of the simulation methodology. Nevertheless, I acknowledge the novelty of this approach and recommend the manuscript for publication in Nat. Comm., provided the following points below have been sufficiently addresses.

Q1. The authors demonstrate clearly that their approach is able to prepare stable and macroscopic thin films of perfectly aligned crystalline blue phase materials that are so far unsurpassed in size (area). But I find the title (ideal single-crystals) and some text in the main body of the manuscript rather misleading in the sense that cubic blue phases are truly three-dimensional materials.

R: We agree with the Referee in that blue phases are truly ordered 3D materials. The purpose of using the term “ideal single-crystal” is meant to emphasize that the proposed strategy allows one to form BP’s that are free of any grain boundaries or lattice defects. We do not know how to better express this concept; in the literature, the term “mono-domain” blue phases is used to denote polycrystalline materials. In our work, we have prepared a single, uninterrupted crystal. We would of course be receptive to any other recommendation that highlights this point.

Q2. Three-dimensional mono-crystalline blue phase I droplets in equilibrium with the isotropic phase have been reported in P.E. Claris, P. Pieranski, M. Joanicot, Phys. Rev. Lett. 52, 542 (1984). Ref. [1] states that faceted mono-crystals of up to 0.1-0.2 mm in size have been grown. The current film thicknesses of 3.5 micrometers or roughly 20 unit cells is an order of magnitude smaller than the typical grain sizes in polycrystalline blue phases. Hence, the authors need to point out more clearly that they are in fact growing a macroscopic thin film, and not a single ideal crystal. It remains to be seen whether their strategy works as well for thicker samples that may be called a single crystal.

R: The Referee is correct; in this work, we report that our patterns enable formation of a uniform and ordered BP over macroscopic areas with a film thickness of approximately 23 BP-unit cells. Even though our BP samples are thin films, the three spatial dimensions of the confined blue phases are significantly longer than the characteristic coherence length ($\xi \sim 10$ nm); therefore, the system can be considered to be a 3D material. Channel thicknesses on the order of a few hundred nanometers can alter the 3D structure of the BP considerably (J. Fukuda and S. Zumer, Phys. Rev. Lett. 104, 017801 (2010)); in our work, however, the channel thickness, in simulations and experiments, is thick enough to avoid such structural changes and distortions in the crystalline symmetry of the BP. In a recent study, we have examined the influence of the channel thickness (and other variables) that influence the formation of ordered crystal BP-samples. That systematic study involved large amounts of data that will be reported in a separate manuscript. Nevertheless, we want to provide the Referee with an image of a 24 μm -thick BPII₍₁₀₀₎ sample (approximately 160 BP-unit cells of thickness) that serves to reinforce the fact that we can prepare truly macroscopic, thick samples.

Q3. Ref. [1] and [2] have gone down into the annals of blue phase research. Regarding the structure of blue phase III, there are more appropriate and up-to-date references, which should be also cited, e.g. S.S. Gandhi, M.S. Kim, J.-Y. Hwang, L.-C. Chien, *Adv. Mater.* 28, 8998-9005 (2016) and O. Henrich, K. Stratford, M.E. Cates, D. Marenduzzo, *Phys. Rev. Lett.* 106, 107801 (2011).

R: We acknowledge the reviewer for this observation. We have included those references in the manuscript.

Q4. In the section 'Pattern Templates' second paragraph the authors say '... A value of $S=1$ corresponds to a materials that is perfectly PARALLEL to the surface, whereas $S=0$ corresponds to an isotropic, disordered region ...' But further above the authors say the surface has been treated for homeotropic anchoring. So it should read 'perpendicular to the surface' rather than 'parallel' in the above sentence.

R: We thank the Referee for pointing out this mistake, The sentence should read “perpendicular”. We have corrected this mistake.

Q5. In the section 'Optimal Pattern Designs' second paragraph the authors mention that the surface energies associated with different BP orientations are similar for all three cases (striped, hexagonal rectangles and circles). Is this because they study a strong anchoring regime? Please add a short explanation to this statement.

R: The Referee is correct: the reason for such behavior in the surface free energy is the strong anchoring condition. We have included the following sentence in the revised version:

“It is important to note that, because of the strong anchoring conditions imposed by the patterned regions, the surface energies associated with different BP orientations are similar for all the three cases considered above”

Q6. Ref. [21] seems to be incorrect or garbled with another reference.

R: The Referee is right: we have moved Ref. 21 to Ref. 20.

Q7. In the section 'Methods' I am missing a term in the surface free energy for planar degenerate anchoring. From Ref. [41] Eq. 3 it is evident that the trace of the square of \tilde{Q} has to be the square of the scalar order parameter $S_0 > 0$. This condition is not met without a second term as in Eq. 4 of Ref. [41].

R: The Referee is correct: we have included the missing term.

Q8. This work contains a substantial amount of modelling and simulation. Whilst I agree that real-world SI units are more important, I would like to see also a parameter mapping from simulation units to real-world units or at least an overview of the simulation parameters that were used in this work.

R: We agree with the Referee. A mapping between simulation parameters and SI units is useful and provides clarity for general readers. We have included such a mapping in the Methods section.

Reviewer #3 (Remarks to the Author):

This paper reports a new strategy to produce large monodomain blue phase samples. The strategy is to prepare substrates with patterned anchoring interaction. The pattern is based on a free energy minimization strategy, and is verified by simulations. Experiments are carried out, producing samples in agreement with predictions. The work is novel, interesting and the results will likely be useful for eventual device applications. I am not altogether happy with the manuscripts; I think the work is better than the manuscript reflects. Improvements need to be made both in organization and style.

***Q1.** Should the paper not begin with an introduction, then perhaps theory, simulations and experiment? The current description is rambling, and the language is unclear. What exactly does it mean that ‘phases reflect light selectively’? What are external ‘cues’? Do the authors really ‘exert control over different planes’? ‘anchoring... induces molecules to lie down’? ‘It can be appreciated from the figure that the sum of Is equal to the lattice size’. ‘Lattice orientation is mediated by the strain...’ ‘theory for the local orientation tensor..’ And so on. The style obscures the physics and the contributions made by the authors; this badly needs to be rectified.*

R: We apologize for the ambiguities and lack of precision in our manuscript. We have attempted to improve the clarity and style of the writing.

***Q2.** There are some scientific questions that need to be addressed as well. The assertion is made that ‘it has not been possible to create macroscopic specimens ..’ followed by the claim that the proposed method make samples can be made over ‘arbitrarily large macroscopic areas’. Some number are needed here. It is not uncommon to see domains of ~ 100 microns; the manuscript reports domains of 200 microns. I understand about control, but there needs to be some comparison of what is now available, and what is realistically possible with the new method. The second issue is thickness. I understand 2.5 microns for simulations, but 3.5 microns is thin. What are device requirements? How far does the induced structure extend into the bulk in thicker samples?*

R: The size of the single-crystal of BPs depends on how large the patterned area is. In this work, we produced 440 μm x 440 μm pattern areas to stabilize BPs. We have included a paragraph in the Discussion section to indicate that the technique employed here to prepare chemically-patterned surfaces has been previously used to pattern areas as large as 300 mm wafers, as it has done on the directed self-assembly of block copolymer (J. Photopolym. Sci. Technol., 26, 831-839 (2013)). On the other hand, as we mention in the second response to the second reviewer, it is possible to produce thicker BP-single crystal samples (we have tried up to 24 μm -thick BP_{II(100)} so far).

Q3. In crystallization, the rate of cooling is usually key in determining domain size. I found no mention of cooling rate, or indeed of any thermal processing, in the manuscript. Other key contributors are thermal gradients – again not discussed.

R: The Referee raises an important observation: we have used a heating process to control the BP crystal growth in this work. We apologize for the lack of information in the Methods section. We have included the following paragraph to clarify this point.

“We start the experiments by heating a cholesteric phase from 25 °C to 39.6 °C. During this process, no visible changes were observed. A heating rate of 0.5 °C/min was then used to reach 39.6 °C. At this point, we used a slower heating rate to increase the temperature by 0.2 °C every 3 minutes.”

The BP-crystal growth based on different thermal processing will be systematically discussed in a separate manuscript.

Q4. The scalar order parameter S is not defined. S=1 indicates a ‘material that is perfectly parallel to the surface’ – what does that mean? The director is parallel? The molecules are parallel (isn’t that S=-1/2)? Without a clear definition, the whole business of S-maps is murky. Again, the work is far better than the manuscript indicates. I urge the authors to strive for clarity and make revisions as indicated.

R: We have corrected these mistakes and added Supplementary Figures S2-S5 to complement our explanation of S-maps and their interpretation.

We thank the Referee for all her/his valuable comments.

REVIEWERS' COMMENTS:

Reviewer #1 (Remarks to the Author):

For the response to my 1st question, it is interesting for me to see how the symmetry of the S-maps remains essentially unaffected by the channel thickness of BP II as shown in Figs S2-S3. Now I agree to author's argument. On the other hand, this insensitivity of S-map to the thickness may not be applicable to BP I because the disclinations of BP I are rectilinear figure and also the S-map at the surface must be different depending on whether, for example, the surface parallel to (110) comprises the lines of disclination or not. If so, the findings of this paper are limited to BP II. Is it true? Why do the authors present only the results on BP II?

Reviewer #2 (Remarks to the Author):

From my point of view the authors have clarified all unclear points that I raised and I recommend this manuscript for publication in Nat. Comm. in its current form.

Reviewer #3 (Remarks to the Author):

The authors have satisfactorily addressed the concerns I raised in my report. The manuscript has been substantially improved, in my opinion, it can now proceed to publication.

Reviewers' comments:

Reviewer #1 (Remarks to the Author):

For the response to my 1st question, it is interesting for me to see how the symmetry of the S-maps remains essentially unaffected by the channel thickness of BP II as shown in Figs S2-S3. Now I agree to author's argument.

Q1. *On the other hand, this insensitivity of S-map to the thickness may not be applicable to BP I because the disclinations of BP I are rectilinear figure and also the S-map at the surface must be different depending on whether, for example, the surface parallel to (110) comprises the lines of disclination or not. If so, the findings of this paper are limited to BP II. Is it true? Why do the authors present only the results on BP II?*

R. We understand the referee's concerns. Below, we include simulation results for BPI with (110)-lattice orientation, confined between two homeotropic surfaces for different film thicknesses. We can observe from these results that the S-map changes, as the reviewer anticipated; however, at least one of the interfaces shows the same behavior –a hexagonal array of squares that favors a homootropic alignment. Figure 6 of the manuscript shows experimental results where a pattern template, with the symmetry of the appropriate S-map, can be used to produce a BPI-single crystal with a (110)-lattice orientation in a 3.5 μm -thick film.

Additionally, we performed experiments using 6.0 μm -thick films to provide additional evidence about how the same pattern works for different film's thicknesses (see figure below).

Finally, in the manuscript we mainly focus on the BPII results because they already involve a significant amount of data analysis, and BPII shows light reflection from all the crystallographic planes. However, we recognize that results for BPI are important to further substantiate the argument that our design strategy can also be applied to the BPI; for this reason, we performed the experiments shown in figure 6 of the manuscript. We understand that additional simulations and experiments were necessary and we thank the Referee for pointing out this observation which we have now fully addressed.

Reviewer #2 (Remarks to the Author):

From my point of view the authors have clarified all unclear points that I raised and I recommend this manuscript for publication in Nat. Comm. in its current form.

Reviewer #3 (Remarks to the Author):

The authors have satisfactorily addressed the concerns I raised in my report. The manuscript has been substantially improved, in my opinion, it can now proceed to publication.